# Physiological and Gene Expression Response of Interspecific Hybrids of *Fraxinus mandshurica* × *Fraxinus americana* to MJ or SNP under Drought

Yang Cao [1,2,†], Liming He [1,2,†], Fei Song [1,2], Chuanzhou Li [1,2], Qitian Ji [1,2], Jianfei Liu [1,2], Guangzhou Peng [1,2], Boyao Li [1,2], Fansuo Zeng [1,2,3] and Yaguang Zhan [1,2,3,*]

1   National Key Laboratory of Tree Genetics and Breeding, Northeast Forestry University, Harbin 150040, China; caoyang2021@163.com (Y.C.); heliming1989111@126.com (L.H.); a390753475@163.com (F.S.); a985197852@163.com (C.L.); strivfe@163.com (Q.J.); liu964278020@163.com (J.L.); peng_guang_zhou@163.com (G.P.); LiboJulius@163.com (B.L.); zengfansuo@126.com (F.Z.)
2   College of Life Sciences, Northeast Forestry University, Harbin 150040, China
3   Heilongjiang Touyan Innovation Team Program, Tree Genetics and Breeding Innovation, Northeast Forestry University, Harbin 150040, China
*   Correspondence: zhanyaguang2014@126.com
†   These authors contributed equally to this work.

**Abstract:** Drought affects the growth and production of *Fraxinus* tree species, such as the precious woody plant *Fraxinus mandshurica*. Based on interspecific hybrid F1 combinations, D110 plants of *F. mandshurica* × *F. americana* with strong drought resistance were selected for this study. To further reveal their heterosis mechanism under drought, in this study, an analysis was conducted pertaining to the physiological indexes and gene expression of related key gene changes in materials of 5 yr D110 seedlings and their female and male parental controls (D113 and 4–3) in response to drought, as well as to the addition of sodium nitrate (SNP, a donor of nitric oxide) and methyl jasmonate (MJ, a donor of jasmonate) signal molecules after drought. The results showed that under drought stress, hybrid D110 plants performed significantly better than their parents, especially compared to D113, in plant growth (the plant height growth was 29.48% higher), photosynthesis (the net photosynthetic rate was 38.4% higher), peroxidation (the increase in MDA content was 71.77% lower), defense enzyme activity (SOD and POD activities were 36.63% and 65.58% higher), hormone contents (ABA, IAA and GA were 33.9%~51.2% higher) and gene expression (the *LHY* and *TOC1* rhythmic genes were 131.97%~165.81% higher). When an exogenous additive agent (SNP or MJ) was applied after drought, the negative effects of drought on growth were effectively alleviated (the tree height growth of D110 increased from 22.76% to 22.32% in comparison to drought conditions); meanwhile, the height growth of D110 plants was significantly higher than that of their parents. Further results of physiological indexes and the expression of related key gene changes in response to SNP or MJ also indicated that D110 plants can recover faster from drought than their parents after application of SNP or MJ. This article provides new ideas for revealing the heterosis mechanism of the drought resistance of interspecific F1 hybrids and supplies effective measures for improving drought resistance in *F. mandshurica*.

**Keywords:** hybrids of *Fraxinus mandshurica* × *Fraxinus americana*; drought stress; SNP; MJ; physiological response; circadian gene expression

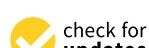



## 1. Introduction

*Fraxinus mandshurica* Rupr., a deciduous tree species belonging to *Fraxinus* Linn., is an excellent and precious tree species with fast growth and high economic value among broad-leaved trees, and it once was recognized as a class II key protected wild plant by the state in 1994 [1]. It is mainly distributed in the arid and semi-arid mountain areas in Northeast China, which consists of the Zhang Guangcai ridge, Daxing'an ridge, Xiaoxing'an ridge

and Changbai Mountains [2]. As one of the main factors restricting the development of forestry, drought not only affects its growth and metabolism, but also affects its yield and reproduction. Therefore, studies on the drought resistance of forestry are of great significance [3]. Exploring the mechanisms of plant drought resistance and finding effective measures to alleviate plant drought stress have important theoretical significance and application value for further cultivating new varieties of drought-resistant *F. mandshurica*.

In the preliminary work, our team obtained interspecific hybrids of *F. mandshurica* × *Fraxinus americana* (another tree species of *Fraxinus* Linn. with high drought resistance), selected the hybrid combination D110 with strong drought resistance [4,5] and studied the advantages of their drought resistance physiological and circadian genes in regulating the drought resistance of hybrid varieties. Studies have shown that the improved hybrid D110 is more adaptive to adversity probably due to the change in the ABA signaling pathway in drought conditions; furthermore, the related change in circadian gene expression may be one of the molecular mechanisms of heterosis [6].

Jasmonic acid (JA) and nitric oxide (NO) are the main signaling molecules in response to plant stress and play significant roles in both biological and abiotic stresses and in plant growth and development. When there is external stimulation, JA has physiological effects such as signaling and initiation of stress resistance gene expression, which participate in not only the regulation of plant growth and development under normal conditions, but also in the response and defense of plants to stress when induced by the environment. Previous studies have showed that exogenous MJ (methyl jasmonate) could improve photosynthetic rate, transpiration and stomatal conductance, enhance antioxidant enzyme activity and reduce malondialdehyde (MDA) content, so as to alleviate plant damage caused by drought stress [7]. NO is involved in plants' responses to a variety of environmental stresses. SNP (sodium nitrate), as a signaling molecule, is a commonly used NO donor. Studies have shown that transgenic *Arabidopsis thaliana* lines transgenic with rat neural NO synthase (nNOS) exhibit high levels of antioxidant enzyme activity, and the photosynthetic capacity, stress tolerance and hormone content of plants are all affected by nNOS transgenic [8–10]. NO is involved in increases in ABA, plant superoxide dismutase (SOD), peroxidase (POD) and ascorbic acid peroxidase activity related to anti-aging [11] and reduces the accumulation of reactive oxygen species under abiotic stress, improving the function of plant resistance to oxidation damage. Moreover, JA can rapidly activate the antioxidant protection system of woody plants, remove reactive oxygen species, reduce the level of oxidative stress and effectively alleviate the damage caused by drought stress [12]. Many studies have shown that low concentrations of MJ can significantly improve the activity of protective enzymes in leaves and enhance plant stress resistance [13]. However, current reports on the improvement of plant drought resistance by exogenous MJ and SNP mainly focus on herbaceous plants, involving few woody plants.

The circadian clock is a kind of timing system, which can provide a mechanism for organisms to adapt to the 24 h day-and-night cycle. In the daily life of plants, the biological clock can enable plants to achieve the corresponding biological activity by activating their related genes at specific time points and inducing synchronous expression. Studies in *Arabidopsis thaliana* have shown that *LHY* (late elongated hypocotyl), *TOC1* (timing of CAB expression 1) and *CCA1* (circadian clock associated) interactions play a vital role in the maintenance of circadian rhythms [14]. As a "big switch", rhythm genes can recognize the regulatory elements upstream of stress signaling pathway genes and regulate plants to respond to abiotic stresses. These rhythm genes have been proven to be key genes in growth and resistance heterosis formation [6] and have great potential for improving growth and drought resistance in tree species, such as *F. mandshurica*.

In order to further explore the drought resistance mechanism of hybrid F1 plants, the physiological response and expression characteristics of genes related to hybrid F1 plants and parental controls to SNP and MJ signal molecules under drought stress were studied in this paper, which provides new ideas for revealing the heterosis mechanism of the drought

resistance of interspecific F1 hybrid plants and supplies effective measures for improving drought resistance in *F. mandshurica*.

## 2. Material and Methods

### 2.1. Plant Material and Water Stress Treatments

5-year-old *F. mandshurica* × *F. americana* F1 hybrid combination (D110) seedlings, open-pollinated progenies from their female parents (D113 as female parental control) and female *F. americana* trees of the same age or older from Beijing botanical garden (4–3 as male parental species control) were subjected to water with holding followed by re-watering in the glasshouse at Northeast Forestry University. The temperature in the glasshouse during the study ranged from 20 °C (night) to 30 °C (day). The plants were planted in plastic buckets with a diameter of 53 cm and a height of 55 cm in a pot in a shed. The planting medium was a mixed soil with fine sand mixed with humus soil in the nursery of the forest farm at a rate of 3:1.

The drought stress was realized by water deficit. The standard of water deficit was set as 25% relative water content [6].

Relative water content (RWC) = Soil actual water content/Soil water holding capacity × 100%

The soil water-holding capacity of experimental soil was about 30%. Each morning and evening, Hydra Probe II Soil Moisture and Salinity Sensors (sdi–12/RS485) were used to control water content at the set level (RWC = 25%), thus maintaining continuous stress for 12 days before rehydration. The growth status were observed and recorded in dry processing and every 3 days after watering. Mature leaves from the same position in plants were collected at 9:00 a.m. and stored in a −80 °C refrigerator after being frozen by liquid nitrogen to study the effects of drought stress and rehydration on the expression of drought-resistant genes in plants.

### 2.2. Exogenous Application of SNP and MJ

The plants with robust and uniform growth were selected, and SNP (as the donor of NO) and MJ (as the donor of JA) solutions with concentrations of 1.0 mmol/L were added at the beginning of irrigation after the drought treatment, and then the materials were selected 12 days after the drought treatment.

### 2.3. Determination of Photosynthesis Parameters

The photosynthetic index was determined by selecting 3 middle leaves of the third pair from the top to the bottom of each plant. The fixed light intensity was 1000 $\mu$mol m$^{-2}$s$^{-1}$, the sample room was 400 $\mu$mol $CO_2$ mol$^{-1}$, the measured leaf area was 6 cm$^2$ and the gas flow rate was 500 $\mu$mols$^{-1}$. Then, the LI-COR 6400XT photosynthetic system (USA) was used to measure the photosynthetic activity of the tested materials before and after the drought treatment and after rehydration. Then, the net photosynthetic rate, intercellular $CO_2$ concentration, stomatal conductance and transpiration rate indicators were recorded. The photosynthetic indexes of each plant were measured three times per leaf [6].

### 2.4. Determination of Peroxidation and Antioxidant Enzyme Activities

For the malondialdehyde (MDA) assay, fresh leaves were homogenized in 5 mL of 100 g L$^{-1}$ trichloroacetic acid containing 250 g L$^{-1}$ thiobarbituric acid and centrifuged at 12,000 rpm for 25 min (4 °C). The mixture was heated to 100 °C for 30 min and then cooled quickly in an ice bath. Subsequently, samples were centrifuged at 12,000 rpm for 10 min (4 °C) and the supernatant absorbance was read at 532 nm. The value for the non-specific absorption at 600 nm was subtracted from the 532 nm reading. The concentration of MDA was calculated using an extinction coefficient of 155 mmol$^{-1}$ cm$^{-1}$ [6].

For extraction of antioxidant enzyme, about 0.5 g of tissue was ground in liquid nitrogen with a pre-cooled pestle and mortar and homogenized in 5 mL of extraction

buffer containing 50 mmol $L^{-1}$ phosphate buffer (pH 7.8) and 1% polyvinylpyrrolidone (PVPP). The homogenate was centrifuged at 12,000 rpm for 20 min at 4 °C and the resulting supernatant was collected for enzyme activity measurement. SOD activity was assayed by the photochemical NBT method. One unit of SOD activity was defined as the quantity sufficient to inhibit the reduction of NBT by 50% per min per mg protein. POD activity was estimated from the absorbance change at 470 nm caused by the oxidation of guaiacol. One unit of POD activity was defined to be equivalent to the amount of enzyme required to degrade 0.01 μM of substrate per min per mg protein [6].

## 2.5. Determination of Phytohormones

The phytohormone contents of abscisic acid (ABA), gibberellin (GA), cytokinin (CTK) and auxin (IAA) in the leaves of *F. mandshurica* × *F. americana* F1 hybrid combination (D110) plants and those of their parents were determined by a Mlbio enzyme-linked immunosorbent assay (Elisa) kit [6].

## 2.6. Real-Time Quantitative RT-PCR

RNA extraction, reverse transcription and PCR amplification of cDNA were performed using a plant RNA extraction kit and DNA scavenging kit from TAKARA. Total RNA was extracted and purified from samples at different stages. RNA was assayed by concentration, electrophoresis and reverse transcription. The expression levels of *LHY*, *TOC1*, *LOX* (Lipoxygenase gene), *NIR* (nitrite reductase gene), *PYR1* (pyracbactin resistance, ABA related), *PP2C1* (protein phosphatase 2C) and *TU* (internal reference) genes were detected simultaneously by fluorescence quantitative PCR of the cDNA (Table 1). Real-time fluorescence quantitative PCR was performed by a fluorescence quantitative PCR instrument, and the methods of $2^{-\Delta\Delta CT}$ were used to calculate the relative expression of these genes.

**Table 1.** Primer sequences used in this test.

| Gene | | Primer Name | Primer Sequence |
|---|---|---|---|
| *LHY* | qPT–PCR | qLHY–F | AGAGGAGGAGCACAATAGGTTT |
| | | qLHY–R | TATGTCCTACTGGAACTCCTTTAAT |
| *TOC1* | qPT–PCR | qTOC1–F | AAGTTGACCTTCCTATGTCTAAA |
| | | qTOC1–R | TTACAATGTCCTTCTCTGCTAGT |
| *LOX* | qPT–PCR | qLOX–F | TGAGTGTGCTTCACCCAATA |
| | | qLOX–R | AAGTGCCTGCTCTGTAAAGTT |
| *NIR* | qPT–PCR | qNIR–F | CCAGTTGGGAACCCTATTG |
| | | qNIR–R | TCGTGGCAGGCATGTATG |
| *PYR1* | qPT–PCR | qPYR1–F | TGGTGGGGAGCATAGATTG |
| | | qPYR1–R | CTTCACAACTGTATCGGCAA |
| *PP2C1* | qPT–PCR | qPP2C1–F | TGGCAGGCTTATTATCTCAA |
| | | qPP2C1–R | TGTTTCTTAGGTGGTGGTTG |
| *TU* | qPT–PCR | qTU–F | AGGACGCTGCCAACAACTTT |
| | | qTU–R | TTGAGGGGAAGGGTAAATAGTG |

## 2.7. Data Processing and Statistics

All physiological and biochemical index data (MDA, antioxidant enzyme activities, qPCR, phytohormone contents) measurements are represented as mean ± standard deviation (n = 3). Effects of drought treatment, biotypes and interactions between the two factors were determined by analysis of variance (ANOVA) followed by Duncan's multiple range test (with a probability level of 0.05 treated as statistically significant) using Excel and SPSS Inc. software (Chicago, IL, USA). The figures were drawn in Prism 7.0.

## 3. Results

### 3.1. Effects of Exogenous SNP or MJ on Relative Growth of Hybrid F1 and Parents under Drought

Through continuous measurement of the plant height (Figure 1A) and ground diameter (Figure 1B) of hybrid combinations treated for 12 days, it was found that drought stress

had a significant inhibitory effect on the growth of plants' height and ground diameter in each group of plants. Under normal and drought stress treatment growth conditions, there were significant difference in plant height growth between hybrid D110 plants and female parental control D113 or male parental control 4–3 plants. Under normal growth conditions, the height growth of hybrid D110 plants is 9.15% and 46.49% higher than that of female parent D113 and male parent 4–3 plants. The hybrid has a faster growth rate than its parents. Under drought stress treatment, the plant height of hybrid D110 was 29.48% and 79.19% higher than that of D113 and 4–3, respectively, showing a significant drought resistance advantage. The diameter growth of hybrid D110 was 43.11% higher than that of D113. At the same time, the addition of exogenous SNP and MJ during drought stress treatment increased the height growth of D110 by 22.76% and 22.32%, respectively. The inhibitory effect of drought stress on plant growth is significantly weakened. After SNP addition under drought stress, the ground diameter growth of D110 was 40.32% higher than that of 4–3. Under drought treatment, the growth of plant diameter significantly exceeded that of the control, which may be a manifestation of *F. mandshurica's* resistance to external stress and D110 can more effectively utilize the stimulation of NO and JA signals to resist external drought stress.

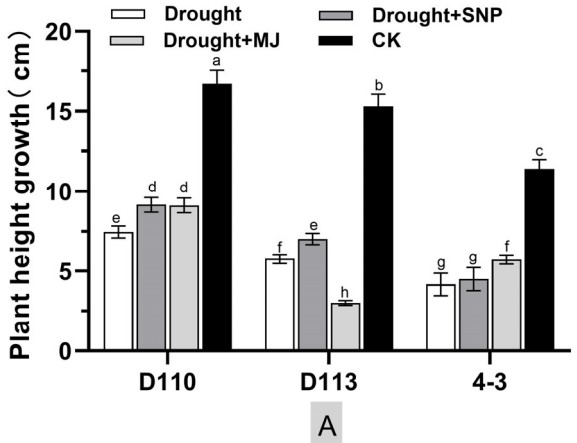 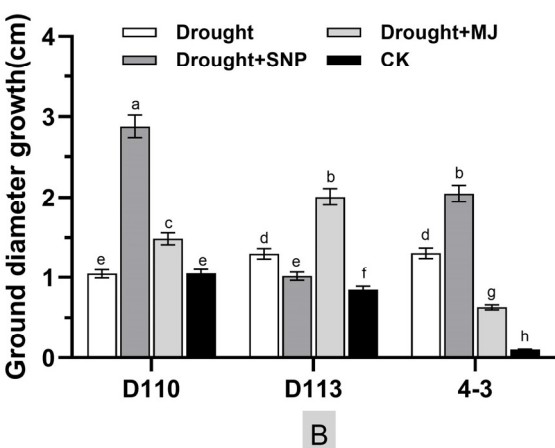

**Figure 1.** The effect of drought stress with exogenous MJ or SNP addition on relative growth. Note: Height of both treatment and control groups were measured after drought stress treatment. (**A**): The plant height growth; (**B**): The ground diameter growth. The D110 is the *F. mandshurica* × *F. americana* F1 hybrid combination, the D113 is the open-pollinated progenies from its female parents as female parental control and the 4–3 represents female *F. americana* trees of the same age or older from Beijing botanical garden used as male parental species control. Meanwhile, Drought means drought stress, Drought + SNP means drought stress by exogenous SNP, Drought + MJ means drought stress by exogenous MJ, CK means control without drought stress. Each data point represents the average of three replicates, and three seedlings were used for each experiment. Error bars represent mean ± SE. Different lowercase letters indicate significant differences at *p* < 0.05. The same as below.

### 3.2. Effects of Exogenous SNP or MJ on Photosynthesis of Hybrid F1 and Parents under Drought

Photosynthesis is one of the basic physiological activities of plants, and it is widely believed that varieties with strong drought resistance can maintain relatively high photosynthetic rates or net photosynthetic rates. Therefore, net photosynthetic rate is a reliable indicator for drought resistance identification. Under drought conditions, the photosynthesis of plants is inhibited and the photosynthetic rate decreases. By measuring the photosynthetic indicators of D110, D113 and 4–3 after 12 days of drought stress treatment, it was found that drought stress inhibited the net photosynthetic rate of D110, D113 and 4–3 to varying degrees and the net photosynthetic rate of D110 under drought stress increased by 38.4% and 50.9% compared to D113 and 4–3 (Figure 2). At the same time, after SNP and MJ addition, the net photosynthetic rate (Figure 2) of D110 exceeded that of its parents D113 and 4–3 by 69.45% and 87.11%, and by 30.69% and 109.13%. The D110 has a higher net

photosynthetic rate and exhibited an advantage in drought resistance. After the application of SNP and MJ, the net photosynthetic rate increased compared to that under drought stress. The application of exogenous substances NO and JA effectively alleviated the damage to plant leaves caused by drought stress.

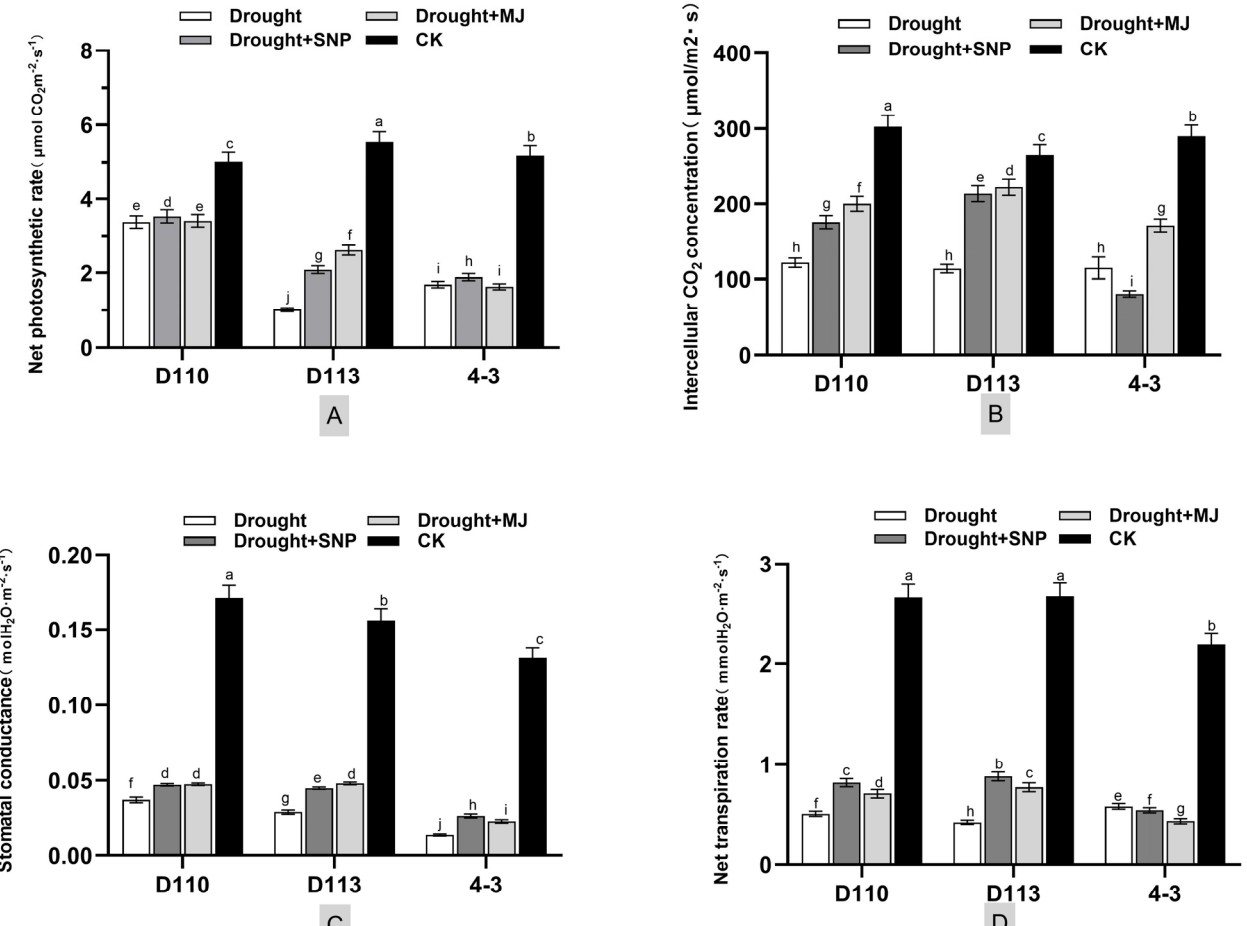

**Figure 2.** The effect of drought stress with exogenous MJ (methyl jasmonate) or SNP (sodium nitrate) addition on photosynthetic parameters. Note: Four indexes, the net photosynthetic rate ($P_n$), stomatal conductance ($g_s$), intercellular $CO_2$ concentration ($C_i$) and net transpiration rate (E) of both treatment and control groups were measured by LI–6400XT after drought stress with exogenous SNP and MJ. (**A**): the $P_n$ index; (**B**): the $C_i$ index; (**C**): the $g_s$ index; (**D**): the E index.

When exogenous SNP or MJ was added, the net photosynthetic rate of the hybrid D110 was 4.51% and 0.92% higher than the values under drought stress (Figure 2), indicating that the sensitivity of the hybrid D110 to photosynthesis can be improved by applying SNP signals through external sources. Overall, the hybrid D110 plants had stronger resistance compared to their parents under drought stress, and the net photosynthetic rate decreased to varying degrees when exogenous SNP or MJ was added compared to the control group. Moreover, applying SNP can allow the plant to recover its photosynthetic rate more effectively under drought stress.

### 3.3. Effects of Exogenous SNP or MJ on the Content of Peroxidation Levels and Antioxidant Enzyme Activities of Hybrid F1 and Parents under Drought

The damage suffered by plants under stress is closely related to membrane lipid peroxidation induced by the accumulation of reactive oxygen species. Malondialdehyde (MDA) is one of the most important products of membrane lipid peroxidation and can bind or cross-link with proteins and enzymes on the cell membrane to inactivate them, thereby

destroying the structure and function of biological membranes. Therefore, the degree of membrane lipid peroxidation can be indirectly measured by measuring the content of MDA, thereby measuring the degree of membrane system damage and plant stress resistance. At the same time, when plants are in harsh environments, a large number of reactive oxygen species accumulate in the plant, leading to damage to the membrane system and affecting normal plant growth. The determination of superoxide dismutase (SOD) and peroxidase (POD) activities can indirectly indicate the degree of oxidative damage in plants, thereby demonstrating their drought resistance ability.

By measuring the MDA content of D110, D113 and 4–3 under drought stress and exogenous application of SNP and MJ (Figure 3), it was found that the MDA content of D110, D113 and 4–3 increased to varying degrees under drought stress, indicating that drought stress caused varying degrees of damage to the membrane system of different plants, while the MDA increase in D110 with stronger drought resistance was lower than that of D113 (by 71.77%). This indicates that the damage to the membrane system of hybrid plants is lower, and the MDA content of hybrid D110 plants is more stable than parents D113 and 4–3. Under drought stress and exogenous application of SNP and MJ, the activities of antioxidant enzymes POD (Figure 4A) and SOD (Figure 4B) were significantly increased compared to the CK group. The amplitude and speed of the increase in POD and SOD enzyme activities in hybrid D110 plants were higher than D113 by 65.58% and 36.63%, indicating that D110 plants had stronger resistance to drought environments than their parents.

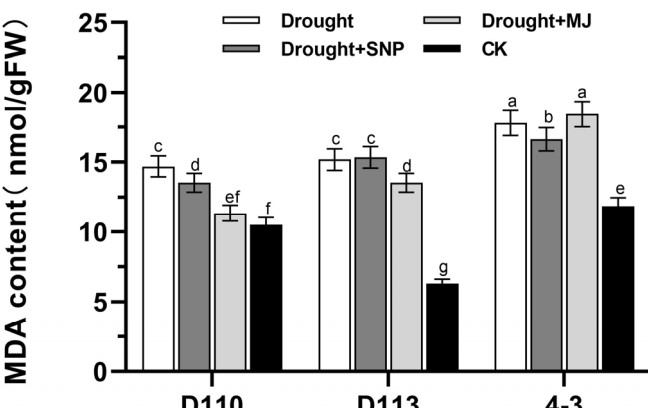

**Figure 3.** The effect of drought stress with exogenous MJ or SNP addition on MDA (malondialdehyde) contents.

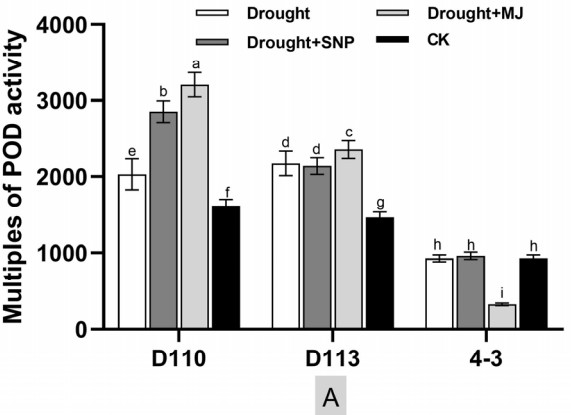
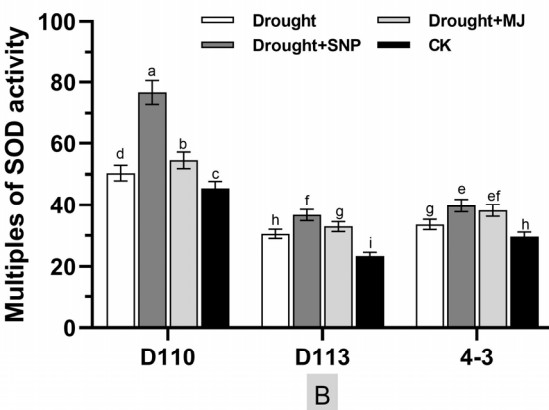

**Figure 4.** The effect of drought stress with exogenous MJ or SNP addition on antioxidant enzyme activities. Note: The SOD (superoxide dismutase) and POD (peroxidase) activities of both treatment and CK groups were measured after drought stress by exogenous SNP and MJ. (**A**): POD activities; (**B**): SOD activities.

After 12 days of SNP and MJ treatment, the enzyme activity of POD in D110 increased by 40.54% and 57.89%, compared to drought stress, and the enzyme activity of SOD increased by 52.27% and 8.18%. Moreover, the application of exogenous hormones can alter the changes in MDA, SOD and POD content, indicating that NO and JA have certain alleviating effects on drought stress. The hybrid D110 plants are more sensitive to exogenous SNP and MJ, and their drought resistance is stronger compared to that of their parents.

### 3.4. Effects of Exogenous SNP or MJ on the Gene Expression of Hybrid F1 and Parents under Drought

As a "big switch", rhythm genes can recognize the original regulation of genes upstream of the stress signaling pathway and regulate the response of plants to abiotic stress. The determination of gene expression can reflect the drought resistance of plants at the molecular level.

After drought stress and exogenous SNP and MJ treatment, the expression of rhythm genes *FmLHY*, *FmTOC1*, *FmLOX* abd *FmNIR* and ABA-related genes *FmPYR1* and *FmPP2C1* was determined in hybrid D110 plants and their parents.

It can be seen that the expression levels of various genes were determined after 12 days of drought stress. The gene expression in the experimental drought treatment group was the lowest in all treatments (Figure 5). Relative gene expression levels of *FmLHY*, *FmTOC1*, *FmLOX*, *FmNIR*, *FmPYR1* and *FmPP2C1* in hybrid D110 plants were 165.81%, 131.97%, 60.56%, 268.34%, 81.11% and 36.44% higher than those of their female parent. This preliminarily result indicates that in drought-resistant hybrid varieties, the expression of rhythm genes is improved to enhance their resistance. The difference in the expression of genes between hybrids and their parents is the reason for the different physiological conditions of hybrids under drought conditions, and also provides a new idea for follow-up studies on the mechanism of the drought resistance advantage of hybrids.

The expression of various genes in the experimental group treated with exogenous SNP and MJ was higher than that in the drought-only treatment group, indicating that exogenous SNP and MJ can increase gene expression, indicating that exogenous hormones have a certain effect on the improvement of plant drought resistance. At the same time, the expression level of the same gene in hybrid D110 plants was higher than that in parent D113 plants; specifically, the LHY gene was 81.09% higher and the TOC1 gene was 46.04% higher, indicating that the hybrid has a much greater drought resistance advantage. When SNP and MJ were applied externally, the expression levels of *LHY* and *TOC1* genes were significantly higher than those under drought conditions.

### 3.5. Effects of Exogenous SNP or MJ on Hormone Contents of Hybrid F1 and Parents under Drought

Plant hormones are organic compounds that naturally exist in plants. Even at low concentrations, plant hormones can coordinate a wide range of physiological processes, including growth and development, as well as responses to abiotic and biological stresses.

Results of IAA, ABA and GA contents were similar. After 12 days of treatment with exogenous SNP and MJ under drought stress, ABA, IAA and GA (Figure 6A,C,D) in hybrid D110 plants showed a significant increase compared to the CK group. Among them, the increase in ABA content was the most significant, with an average increase of 37.9%, 33.9% and 51.2% compared to the CK group. However, after 12 days of treatment with exogenous SNP and MJ under drought stress, the ABA content in parent D113 plants increased by an average of 3.3% and 13.1%, respectively, compared to the CK group: 0.9%. The ABA content in hybrid D110 plants was much higher than that of parent D113 plants, indicating that the hybrids can respond to drought stress stronger and faster at the hormone level and have drought resistance advantages.

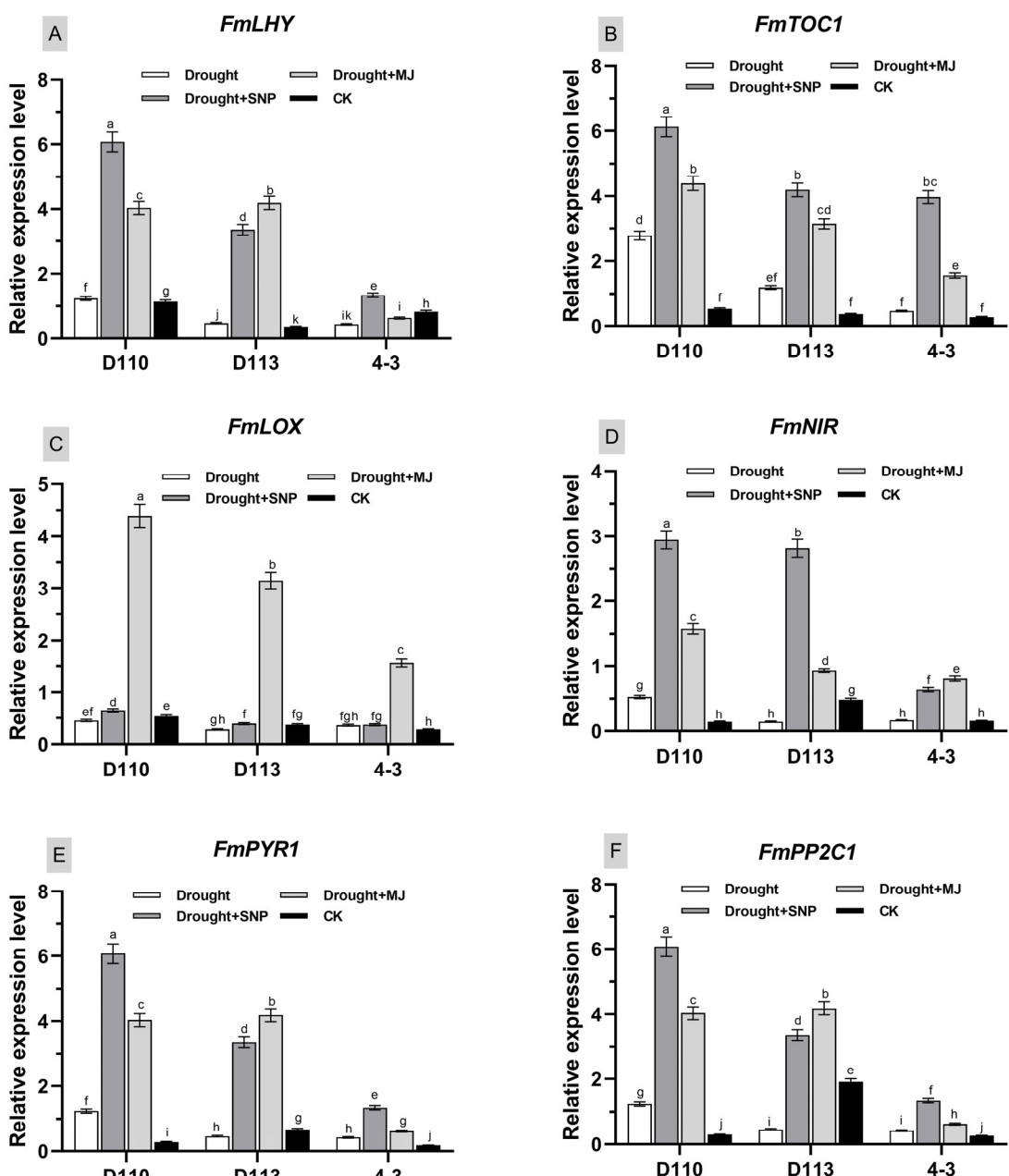

**Figure 5.** The effect of drought stress with exogenous SNP or MJ application on expressions of circadian clock genes and hormone-related genes. Note: The expressions of circadian clock genes *FmLHY* (**A**) and *FmTOC1* (**B**); MJ-related genes *FmLOX* (**C**); SNP-related genes *FmNIR* (**D**); ABA-related genes *FmPYR1* (**E**) and *FmPP2C1* (**F**) of both treatment and CK groups were analyzed by quantitative RT–PCR (a–Tubulin as a control) during daytime (ZT6, 9:00) after drought stress.

Cytokinin is a plant hormone that can modify root morphology and increase root biomass by reducing the ratio of root crown to hypocotyl (reducing CTK levels) to enhance plant drought resistance. The concentration of CTK (Figure 6B) showed a decreasing trend under drought stress and was significantly lower than that of the CK group. After 12 days of drought treatment, the CTK content of hybrid D110 plants decreased by 10.4% compared to the CK group, and that of the parent D113 plants decreased by 8% compared to the control group. It was proven that hybrid D110 plants could respond faster and stronger to drought stress and enhance their survival rate in arid environments by adjusting hormone contents. The difference in hormone changes between hybrids and parents may also be one of the reasons why hybrids have drought resistance advantages.

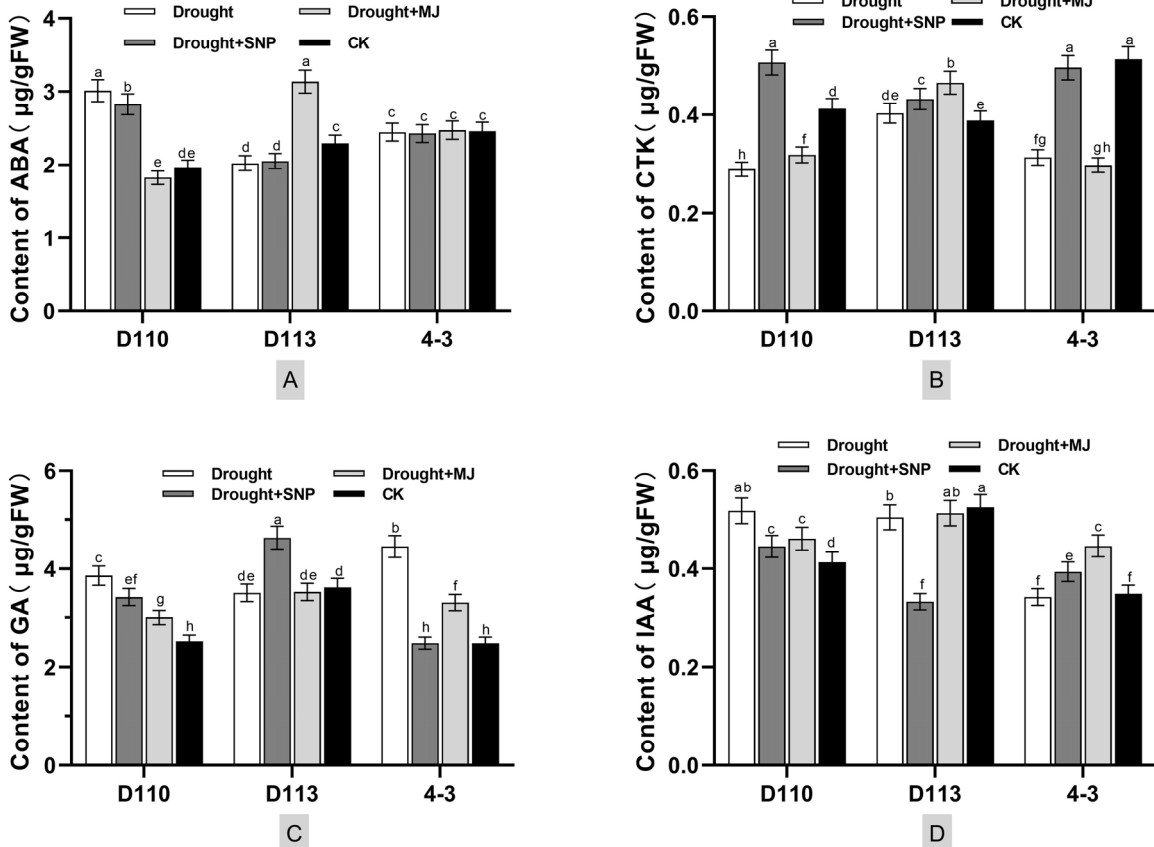

**Figure 6.** The effect of drought stress by exogenous SNP or MJ on endogenous phytohormone contents. Note: The ABA (**A**), CTK (**B**), GA (**C**) and IAA (**D**) contents of both treatment and CK groups were measured after drought stress with exogenous SNP and MJ application.

## 4. Discussion

Drought is an environmental stress that has a serious impact on seed germination, plant growth and productivity. Plants with strong drought resistance are ideal candidates for studying drought-related genes, proteins and metabolites. In past studies, it has been found that hybrids can enhance the expression of rhythmic genes to enhance the heterosis of drought resistance [6]. In this study, we subjected interspecific hybrid D110 seedlings and dual parental controls (D113 and 4–3) to drought stress, while applying exogenous SNP or MJ, and then their physiological indicators were measured to search for reasons for the drought resistance of D110 hybrids and to provide a theoretical basis for future breeding in *Fraxinus* species, such as *F. mandshurica*.

The growth rate of D110 hybrids was significantly higher than that of D113 and 4–3 plants under normal conditions and drought stress. Under drought stress, the growth of D110 hybrids was inhibited, but the degree of inhibition was significantly lower than that of its parents D113 and 4–3, reflecting the better drought resistance ability of the hybrid plants. To underline the reasons behind the growth and drought resistance heterosis of hybrid F1, photosynthetic physiological indexes were first studied.

Photosynthesis is jointly controlled by stomatal limiting factors and non-stomatal limiting factors. Stomata promote the opening and closing of $CO_2/H_2O$ exchange, and its switch is mainly related to the morphology of guard cell, auxiliary cells and epidermal cells [15]. The non-stomatal limiting factor mainly refers to the assimilation reaction of $CO_2$, which is responsible for converting light energy into chemical energy. FARQUHAR proposed a consistent trend between intercellular $CO_2$ concentration and net photosynthetic rate, which can be used to explain why stomatal limiting factors are the main cause of

changes in photosynthesis; otherwise, it is caused by non-stomatal limiting factors, that is, changes in the photosynthetic activity of mesophyll cells [16].

Under the experimental conditions of this study, under drought treatment, the net photosynthetic rate (Pn), transpiration rate (E), stomatal conductance (Gs) and intercellular $CO_2$ concentration (Ci) of D110 plants and their parents decreased, indicating that stomatal limitations are the main factor affecting plant photosynthesis. Drought stress leads to partial or complete closure of stomatal movement in hybrid cells, reducing the ability of mesophyll cells to utilize carbon dioxide and also reducing $CO_2$ fixation. Under drought conditions, the intercellular $CO_2$ concentration, stomatal conductance and transpiration rate of D110 plants increased compared to that of D113 and 4–3 plants. This indicates that D110 hybrids have better resistance to stomatal limitations and enhance the water retention ability of plant tissues. This result is consistent with previous studies [17]. The increase in the net photosynthetic rate may also be one of the reasons why hybrid varieties have drought resistance advantages. After applying SNP and MJ, the net photosynthetic rate (Pn), transpiration rate (E), stomatal conductance (Gs) and intercellular $CO_2$ concentration (Ci) of D110 plants increased compared to drought conditions, indicating that applying SNP and MJ can promote stomatal opening in *Fraxinus* leaves and, to some extent, alleviate the damage caused by drought. The reason may be that JA can reduce water loss under transpiration and prevent excessive water loss of leaves by regulating the stomatal aperture to induce stomatal closure, thus enhancing the ability of tissues to maintain water [18]. The net photosynthetic rate, stomatal conductance and intercellular carbon dioxide concentration of *Fraxinus* were increased after the application of exogenous SNP, possibly due to the promotion of hydrogen sulfide signal generation by exogenous SNP. Hydrogen sulfide can promote an increase in net photosynthetic rate by increasing stomatal opening and density [19]. On the other hand, exogenous SNP can also increase the net photosynthetic rate of plants by increasing the content of photosynthetic pigments and increasing the opening ratio of PS II reaction centers [20].

Overall, the D110 hybrids had stronger resistance compared totheir parents under drought stress, and the net photosynthetic rate decreased to varying degrees when exogenous SNP or MJ was added compared to the control group. However, the magnitude of the decrease was lower than the photosynthetic rate under drought conditions. Moreover, applying SNP can enable the plant to recover its photosynthetic rate more effectively under drought stress.

When exogenous SNP and MJ were applied, the enzyme activities of SOD and POD in *Fraxinus* under drought stress can be increased. With the extension of drought stress time, the ability of plants to overcome oxidative stress depends on the activity of antioxidant enzymes such as SOD and POD. As an oxygen free radical absorber, SOD's main function is to decompose reactive oxygen groups and slow down the rate of membrane lipid peroxidation. Plants with strong drought resistance have stronger SOD activity. POD can disproportionate $H_2O_2$ produced by SOD into $H_2O$ and oxygen molecules to maintain low levels of reactive oxygen species, thereby improving the survival rate of plants in arid environments. At the same time, cultivars with higher drought tolerance have lower MDA content under stress [21]. Under drought stress, the SOD and POD enzyme activities of D110 hybrids were significantly higher than those of parent D113 plants, indicating that D110 hybrids can reduce the degree of stress-induced damage to plant membranes by increasing enzyme activity, achieving significant stress resistance. D110 hybrids were more effective in reducing MDA production than their parents, and increased the content of SOD and POD, thereby resisting external environmental changes. When subjected to drought treatment and exogenous application of SNP and MJ, the MDA content of each combination was lower than that under drought stress alone. In addition, the results showed that applying MJ could more effectively inhibit the production of membrane lipid peroxide MDA than applying SNP.

Plants can produce a large number of free amino acids and polyunsaturated fatty acids (PUFA) because the decomposition rate of proteins and lipids is greater than the synthesis

rate in arid environments, and a large number of metabolites produced by the oxidation of PUFA are collectively called oxidized lipids (such as jasmonic acid, JA). Many of them are bioactive compounds that may directly participate in plant defense, communication with other organisms or signal transduction in plant development, defense responses and stress adaptation [22–24]. The substrate of lipoxygenase is a free polyunsaturated fatty acid (PUFA) with increased content under stress conditions. Exogenous ABA can stimulate the activity of lipoxygenase and promote the production of JA, while JA combines with amino acid isoleucine or is methylated to produce methyl jasmonic acid, MJ. Activating the peroxide oxidation of membrane lipids promotes the formation of tolerance in damaged rice leaves [17]. Therefore, an increase in ABA concentration under drought stress is beneficial for improving plant survival rate.

Abscisic acid (ABA) plays a crucial role in the response of plants to drought stress. ABA mediates drought response by regulating the expression of drought response genes and stomatal closure. ABA regulates signal transduction pathways and gene expression levels through rapid induction or inhibition of transcription [25]. The plant hormone abscisic acid (ABA) plays an important role in many physiological processes, such as seed dormancy, seed germination, seed growth and response to biotic and abiotic stresses. One of the functions of ABA in addressing adverse environmental conditions is to regulate the movement of stomata [26]. Auxin (IAA) is a kind of chemical signal substance that exists in all vascular plants. IAA can regulate many aspects of plant growth and development, including cell division and elongation, as well as organ development at cellular and plant levels [27]. This physiological regulation is achieved through signal transduction causing changes in the expression levels of many genes. Gibberellin (GA) can inhibit cell proliferation by increasing the intensity of cell cycle inhibitors. GA not only inhibits growth, but also promotes plant survival under pressure conditions by limiting the accumulation of reactive oxygen species (ROS), thereby delaying cell death [28].

Under drought stress and exogenous application of SNP and MJ, the expression of three endogenous hormones GA, auxin (IAA) and abscisic acid (ABA) in hybrid D110 plants was significantly higher than that in D113 plants, and the rate and extent of the increase of these three hormones' contents when exogenous MJ was applied to the same plant were greater than those resulting from the exogenous application of SNP. Research has shown that under drought stress, when external MJ and SNP are applied, the rate and amplitude of CTK content reduction in D110 hybrids are lower than that of parental D113 plants, indicating that D110 hybrids still have strong cell division ability and resilience after drought stress, which is conducive to the survival of Fraxinus mandshurica in arid environments [29–34].

We believe that under changes in the external environment, the change in plants' internal hormones is the most intuitive indicator of performance of plants' growth state. By detecting the endogenous hormones of related plants, it was shown that the endogenous hormone response of D110 hybrids was stronger than that of their parents D113 and 4–3, which promoted its growth and improved its stress resistance. When exogenous hormones MJ and SNP were applied, the demand of plants for hormones was alleviated, resulting in changes in the content of endogenous hormones, which alleviated the damage of drought to plants and improved the survival rate of plants. The difference in hormones between hybrids and their parents is one of the reasons why hybrids have drought resistance advantage.

The *LHY* gene and *TOC1* gene, as the core oscillators of the plant biological clock system, together form a transcriptional inhibition loop. The response of the *LHY* gene and *TOC1* gene to drought stress shows an increasing trend in amplitude [35]. Regardless of the presence or absence of exogenous SNP and MJ under drought stress, there is a significant increase in the expression of rhythm genes between D110 and D113. However, the increase in heterozygous D110 was greater than that of D113, and the response speed is also faster, indicating that the drought resistance advantage of hybrids is related to the expression of rhythm genes. *LOX* can synthesize the precursor of MJ and its metabolites are oxygen free radicals and reactive oxygen species which can destroy the membrane

structure of cells and participate in the process of plant senescence. The results showed that *LOX* genes synthesized during germination could increase the expression level through the abscisic acid pathway [36]. It is reported that higher nitrogen uptake can mitigate the destructive effects of drought stress. After drought stress, the nitrogen and proline contents of transgenic plants were higher than that of a wild-type control, and the activity of nitrite reductase (*NIR*) was also higher during nitrate assimilation. The results showed that higher nitrate transport and assimilation activity were helpful to improve drought resistance of transgenic plants [37]. Most of the genes encoding nitrate transporter proteins and enzymes responsible for N assimilation and reactivation (such as nitrite reductase *NIR*) are down-regulated under abiotic stress. Especially under long-term stress (24 h), this may be one of the reasons for the decrease in plant growth and development under abiotic stress [38].

The *LOX* and *TOC1* bases of lipoxygenase genes are important genes in plant response to drought stress, and *LOX* activity is considered a biological marker of plant physiological status. The regulation of MJ can induce *LOX* expression. When applied with exogenous SNP and MJ under drought stress, the expression of *NIR* and *LOX* genes in the hybrid offspring D110 was higher than that in the parental D113. Under drought stress, the exogenous application of MJ led to high expressions of the *LOX* gene, which could quickly open the resistance of plants to external biotic stress and increase their tolerance. *NIR* is mainly involved in the absorption and transformation of N in nutrient synthesis, and the high expression level in hybrid offspring indicates that the hybrids can generate more energy to maintain growth in arid environments. These genes' expression in the D110 hybrids was higher than those of their parents, and the mechanism of the drought-resistant advantage of the hybrids can be explained to some extent; further reasons for the drought-resistant advantage of the hybrids will be disclosed with more evidence in future.

## 5. Conclusions

In summary, there are significant differences in physiological indicators and gene expression between interspecific hybrid D110 plants and their parents (D113 and 4–3) under drought stress and exogenous application of SNP and MJ. Specifically, interspecific hybrid D110 plants can eliminate excess reactive oxygen species by enhancing plant photosynthesis, enhancing enzyme activities such as SOD and POD, and regulating the expression of related hormones to maintain cell membrane integrity. The comprehensive effect of multiple factors bestows interspecific hybrids with greater advantages under drought stress. Moreover, the expressions of rhythm genes *LHY* and *TOC1* and stress-related genes *NIR* and *LOX* in D110 and D113 were different, which can further explain the reasons and mechanisms of drought resistance advantages in interspecific D110 hybrids. The exogenous application of SNP and MJ has varying degrees of improvement in the drought resistance ability of F1 hybrids and their parents. Drought is a natural disaster that occurs irregularly, so studying the drought resistance ability of *Fraxinus* species can provide a theoretical basis for solving the drought stress problem of *Fraxinus* tree species, such as *F. mandshurica*, at various growth stages, and provide a theoretical basis for the planting and yield improvement of *Fraxinus*.

**Author Contributions:** Y.Z., L.H. and Y.C. conceived the idea of the study and designed the experiment. Y.C. and L.H. performed most of these experiments and the C.L., Q.J., J.L., G.P. and B.L. provided their help in treatment and sample collection. Y.C. and F.S. analyzed the data and drew the figures. Y.Z., Y.C., L.H., F.S. and F.Z. were involved in writing and revising the manuscript. All authors approved the final manuscript. All authors have read and agreed to the published version of the manuscript.

**Funding:** This research was supported by the National Key R&D Program of China (2021YFD2200303) and Heilongjiang Province Applied Technology Research and Development Program Key Project (GA19B201). This funding play roles in the design of the study and collection, analysis and in writing the manuscript.

**Data Availability Statement:** All data generated or analyzed during this study are included in this published article. The experimental materials are the seeds obtained in our laboratory, and the complete plants were cultured in the greenhouse of Northeast Forestry University. We abide by relevant regulations.

**Conflicts of Interest:** The authors declare that they have no competing interest.

**Abbreviations**

Drought: drought stress; Drought + SNP: drought stress by exogenous SNP; Drought + MJ: drought stress by exogenous MJ; CK: control without drought stress; SNP: sodium nitrate; MJ: methyl jasmonate; JA: jasmonic acid; NO: nitric oxide; *nNOS*: neural NO synthase; SOD: superoxide dismutase; POD: peroxidase; MDA malondialdehyde; N: nitrogen; PUFA: polyunsaturated fatty acid; ROS: reactive oxygen species; ABA: abscisic acid. *LHY*: late elongated hypocotyl; *TOC1*: timing of CAB expression 1; *CCA1*: circadian clock associated 1; *NIR*: nitrite reductase; *PYR1*: pyracbactin resistance 1; *PP2C1*: protein phosphatase 2C.

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
