# Peer review of "Physiological and Gene Expression Response of Interspecific Hybrids of Fraxinus mandshurica × Fraxinus americana to MJ or SNP under Drought"

_forests, doi:10.3390/f14061277_

Round 1

Reviewer 1 Report

In this manuscript, the authors describe the results of studying the effects of drought and growth stimulants on the growth of ash plants. This experiment provides a new theoretical basis for the selection of Manchurian ash for increased drought resistance in order to increase its survival in nature.

Research aimed at solving these problems is relevant. The uniqueness of the article is sufficient for publication.

Along with the positive aspects of the work, there are also negative ones that need to be corrected.

1. There are grammatical, stylistic and punctuation errors in the text. The word Interspecific is misspelled in the name.

2. The authors write in the title about the hybrid of the Manchurian ash, but in fact, as can be seen from the following text, we are talking about two species: Fraxinus mandshurica and Fraxinus Americana, as well as an interspecific hybrid between them, which cannot be attributed to Fraxinus mandshurica.

3. It is not clear from the results what size (height and diameter) the ash plants were when they were planted in the vessels. How long did they grow up there?

4. In Figure 1, it is not clear what the designation CK, control means? There are no numerical values of morphological features. From the graph, you can only approximately represent the numbers. The ordinate axis shows the increment or absolute values of the features? In graph 1B, the increase in diameter in the D113 sample in the variant of Dropped+SNP – 2.8 cm seems incredible, whereas in the control it is 1.0 cm. At the same time, there was almost no increase in the 4-3 sample at the control.

5. The authors claim that under normal growing conditions there is a little difference in plant height growth between hybrid D110, female parent D113 and male parent 4-3, and when under drought stress treatment, the plant height of hybrid D110 was 29.48% and 79.19% higher than that of parent D113 and the father 4-3. However, it can be seen from the graph that the height difference between plant species was more significant at the control.

 The article is accepted after revision (correction of methodological errors and text editing). After correcting these shortcomings, the article can be published in the journal "Forests".

Author Response

Dear Editors and Reviewers:
Thank you for your letter and for the reviewers’ comments concerning our manuscript entitled “Drought physiology with exogenous application of SNP and MJ and gene expression characteristics of Fraxinus interspecific hybrids” (ID:   forests-2415588 ). Those comments are all valuable and very helpful for revising and improving our paper, as well as the important guiding significance to our researches. We have studied comments carefully and have made correction which we hope meet with approval. Revised portion are marked in red in the paper. The main corrections in the paper and the responds to the reviewer’s comments are as flowing:Please see the attachment

Reviewer 2 Report

The proposed subject is interesting, but the document is largely perfectible. The presentation of the results is not clear and one must note numerous typographical errors in the text. The legends of the figures are incomplete making the analysis of the results difficult.

Introduction deserves more explanation on the strategic choice of the authors to study drought stress through osmotic stress for this species. Some sentences are difficult to understand.

M&M, Here again there are many inaccuracies, many typographical errors and above all a lack of reference as to the methods used. I don't understand why no basic water potential measurements were made to qualify the plants stress. I also lack explanations when choosing the parameters used to make the measurements

Results and discussion, If the processing of the data seems to be correct, the discussion only resumes the results without making any basic physiological analysis. For photosynthesis, it seems that this tree regulates water stress via non-mesophyllian processes. What role MJ and NO on these processes? etc..

In conclusion, this work is interesting, but deserves to be taken up and corrected before it can be published.

Author Response

(The authors gave the same response as above.)
